# Leukocyte Telomeric G-Tail Length Shortening Is Associated with Esophageal Cancer Recurrence

**DOI:** 10.3390/jcm11247385

**Published:** 2022-12-12

**Authors:** Jiayan Han, Soichiro Hayashi, Ryou-u Takahashi, Ryosuke Hirohata, Tomoaki Kurokawa, Mizuki Tashiro, Yuki Yamamoto, Morihito Okada, Hidetoshi Tahara

**Affiliations:** 1Department of Cellular and Molecular Biology, Graduate School of Biomedical and Health Sciences, Hiroshima University, 1-2-3, Kasumi, Minami-ku, Hiroshima 734-8553, Japan; 2Department of Surgical Oncology, Hiroshima University, 1-2-3, Kasumi, Minami-ku, Hiroshima 734-0037, Japan

**Keywords:** esophageal cancer, leukocyte telomere length, telomeric G-tail length, biomarker

## Abstract

Despite significant advances in therapeutics for esophageal cancer (ESC) in the past decade, it remains the sixth most fatal malignancy, with a poor 5-year survival rate (approximately 10%). There is an urgent need to improve the timely diagnosis to aid the prediction of the therapeutic response and prognosis of patients with ESC. The telomeric G-tail plays an important role in the chromosome protection. However, aging and age-related diseases lead to its shortening. Therefore, the G-tail length has been proposed as a novel potential biomarker. In the present study, to examine the possibility of G-tail shortening in patients with ESC, we measured the leukocyte telomere length (LTL) and the G-tail length using a hybridization protection assay in 147 patients with ESC and 170 age-matched healthy controls. We found that the G-tail length in patients with ESC was shorter than that in the healthy controls (*p* = 0.02), while the LTL shortening was not correlated with the ESC incidence and recurrence. Our results suggest that the G-tail length reflects the physiological status of patients with ESC and is a promising biomarker for the diagnosis and prognosis of ESC.

## 1. Introduction

Esophageal cancer (ESC), which is classified into two main types: esophageal squamous cell carcinoma (ESCC) and esophageal adenocarcinoma (EAC), is the eighth most common cancer worldwide and the sixth leading cause of cancer-related deaths [1,2,3]. The global incidence of ESCC accounts for 90% of all ESC cases each year [3,4,5]. Reports have shown that ESCC is highly prevalent in Asia [6].

Due to the lack of early clinical symptoms, ESC is commonly diagnosed at an advanced stage. This is one of the main reasons for the poor 5-year survival rates (approximately 10%) of patients with ESC [7,8]. Patients with advanced-stage cancer are unable to undergo surgery, leading to a poor prognosis. Despite improvements in the therapeutic approaches, such as surgery, chemotherapy, radiotherapy, and chemoradiotherapy (CRT) [6], the prognosis remains poor, with the 5-year post-esophagectomy survival rate, ranging from 15–40% [7].

Existing methods for the diagnosis of ESC include endoscopy, positron emission tomography, biopsy, computerized tomography, and immunohistochemistry [9,10]. The disadvantages of the conventional diagnostic methods include their high cost and invasiveness. Therefore, it is critical to develop biomarkers for ESC that enable the rapid and noninvasive diagnosis at an early stage of the disease.

Telomeres, which contain tandem TTAGGG repeats, are nucleoprotein complexes located at the ends of eukaryotic chromosomes. Their main function is to maintain the genomic stability [11,12]. Telomeres form large loops (called telomeres or t-loops) by the invasion of the G-rich single-stranded overhang (G-tail) into double-stranded telomeric DNA, to prevent the chromosome ends from being recognized as a DNA break [11,12,13]. The telomeric G-tail is vital for the formation of the telomeric loop structure [14]. Telomeric sequences are lost during the successive cell divisions, and DNA damage accelerates the telomere shortening, which eventually results in the genomic destabilization [14,15]. We previously established a high-sensitivity and high-throughput method, known as the G-tail telomere hybridization protection assay (Gt-telomere HPA), to measure the telomeric G-tail length in various types of cells and clinical samples [16]. 

Several clinical studies have shown that the leukocyte telomeric sequence length reflects the length of other cell types in the body, a process known as synchrony [17,18,19]. Consistent with these studies, we reported that the telomere shortening and/or telomeric G-tail shortening is associated with aging and age-related diseases, including white matter lesions, endothelial dysfunction, and cancer [20,21,22,23]. In breast cancer, the telomeric G-tail length is shorter in patients at an early stage than in healthy individuals, suggesting that the telomeric G-tail length is a promising blood-based biomarker for the diagnosis of cancer. Considering our previous findings, in this current study, we investigated whether the shortening of the telomeric G-tail and total telomere correlates with the physiological status of patients with ESC.

## 2. Materials and Methods

### 2.1. Study Subjects

A total of 147 patients with ESC aged 45–88 years (median age 69.0 years, interquartile range (IQR) = 63.0–74.0 years; 122 males and 25 females) were registered at Hiroshima University Hospital, between May 2018 and April 2022. All patients were confirmed to have ESC and were treated, based on the guidelines for the diagnosis and treatment of carcinoma of the esophagus in April 2012, edited by the Japan Esophageal Society. The tumor stage was classified, based on the 8th edition of the Union for International Cancer Control and the American Joint Committee on Cancer. the baseline clinical characteristics, including age, sex, smoking status, and alcohol consumption, were recorded. The inclusion criteria included the following: (1) all patients aged 40–90 years with a primary esophageal cancer, (2) informed consent, (3) no evidence of metastases or recurrence at the initial assessment. The exclusion criteria included the following: (1) patients with concomitant multiple cancers, (2) lost to follow-up, (3) evidence of metastases or recurrence at the initial assessment.

Blood samples were obtained from the volunteers. Out of these volunteers, we selected individuals who were age-matched with our study participants. Finally, we recruited 170 healthy participants aged 40–80 years (median age 62.0 years, IQR = 58.0–68.3 years; 88 males and 82 females) as control subjects. The criteria for selection of healthy volunteers included the following: (1) healthy adult volunteers aged 40–90 years, (2) with no severe underlying disease, (3) informed consent.

This study was approved by the Institutional Research and Ethics Committee of the Hiroshima University Hospital, Hiroshima, Japan. All of the participants provided informed consent to participate in the study.

### 2.2. Measurement of the Leukocyte Telomere Length and Telomeric G-Tail Length

Peripheral blood samples (10 mL) were collected from each participant. The total telomere length was measured in the genomic DNA of the peripheral blood leukocytes by the telomere HPA [24], and the telomeric G-tail length was measured using the telomere G-tail HPA, as previously described [16]. the genomic DNA was extracted using the phenol-chloroform extraction method. The DNA concentration used in the assay was measured using a Nanophotometer Pearl (Implen GmbH, Munich, Germany).

The total telomere length was determined using 0.2 µg of denatured genomic DNA, and a telomeric G-tail length was determined using 1 µg of non-denatured genomic DNA, using a 96-plate automated machine, JANUS Automated Workstation, combined with an EnVision Multilabel Plate Reader (PerkinElmer, Waltham, MA, USA). All samples were assessed in triplicates, and the genomic DNA extracted from the human colon cancer cell line RKO was used as a control to correct for the inter-assay variability. The average coefficient of variance in all samples was 5.7% for the telomeric G-tail length and 4.9% for the total telomere length, whereas the inter-assay coefficient of variance was 5.6% for the telomeric G-tail length and 5.5% for the total telomere length.

### 2.3. Data Analysis

A statistical analysis was performed using GraphPad Prism version 8.4.0 (GraphPad Software, San Diego, CA, USA, www.graphpad.com). The normal distribution of the variables was tested using the D’Agostino–Pearson omnibus normality test and by visual inspection of the QQ plot. The independent sample t-test was performed for the normally distributed continuous variables, and the Mann–Whitney U test was used for the non-normally distributed variables. Data are presented as mean ± standard deviation for the normally distributed variables; median (IQR) for non-normally distributed variables, and as numbers of observations (%) for the categorical variables. For all analyses, the statistical significance of the inter-group differences was set at a level of *p* < 0.05, and was assessed using the unpaired *t*-test, Mann–Whitney U test, one-way ANOVA, Kruskal–Wallis test, and Dunn’s multiple comparisons test, as appropriate. The correlations between the total telomere length, telomeric G-tail length, and other variables were examined using Spearman’s correlation coefficient test and a linear regression analysis.

## 3. Results

### 3.1. Telomeric G-Tail Length Is Associated with the Cancer Status

The clinical characteristics of the controls and patients with ESC are shown (Table 1). We compared the LTL and G-tail lengths in the controls and patients with ESC. Our data showed that the telomeric G-tail length in patients with ESC was shorter than that in the controls (*p* = 0.02), while there was no significant difference in the LTL between controls and patients with ESC (*p* = 0.07) (Figure 1A). 

The median luminescence signals of the LTL in the control participants and patients with ESC were 106,298 relative light units (RLU) (IQR = 96,077–114,652 RLU)/µg DNA and 101,246 RLU (IQR = 92,850–112,002 RLU)/µg DNA, respectively. The median luminescence signals of the G-tail length in the control participants and patients with ESC were 9653 RLU (IQR = 8465–10,775 RLU)/µg DNA and 9015 RLU (IQR = 7662–10,814 RLU)/µg DNA, respectively.

To investigate the determinants of these results, we compared the LTL and the G-tail length in patients with ESC, according to the disease stage. A significant difference was observed only in the G-tail length between the controls and patients with stage II ESC (*p* = 0.01). In contrast, there was no significant difference in the LTL between the controls and patients with ESC at any stage (Figure 1B). 

Additionally, because the tumor-node-metastasis (TNM) classification systems of the Union for International Cancer Control and the American Joint Committee on Cancer are widely used to predict the prognosis of patients with ESC, we compared the LTL and the G-tail length in the controls and ESC patients, according to each parameter. Our data showed that a significant difference was observed only in the G-tail length between the controls and patients with ESC in the T3 category (*p* = 0.03). In contrast, there were no significant differences in the LTL and the G-tail length between the controls and patients with ESC in the N or M category (Appendix A).

### 3.2. Correlation between Aging and the Telomeric G-Tail Length

Next, we investigated the correlations between aging, the LTL, and the G-tail length. Our data showed that the LTL was negatively correlated with age in both controls (r = −0.312, *p* < 0.0001) and patients with ESC (r = −0.287, *p* = 0.0004) (Figure 2). The G-tail length also correlated with age in the controls (r = −0.178, *p* = 0.020) and patients with ESC (r = −0.258, *p* = 0.002) (Figure 2). The trend of the LTL shortening with age among patients with ESC was similar to that among the controls. However, we observed that the trend of the G-tail length shortening among patients with ESC over 50 years of age was more rapid than that among the controls, suggesting that the G-tail shortening was mainly caused by the ESC incidence.

### 3.3. Telomeric G-Tail Length Shortening Correlates with the Recurrence

Next, we compared the LTL and G-tail length in the controls and the preoperative patients with recurrent or non-recurrent ESC, following treatment, to explore the association between the G-tail length and the possibility of recurrence. Our data revealed that there were significant differences in the G-tail length between the controls and the preoperative patients with recurrent ESC (*p* = 0.002), and between the preoperative patients with recurrent ESC and those without recurrent ESC (*p* = 0.02), while no significant difference was observed in the LTL in the same compared groups (*p* = 0.30; *p* = 0.99) (Figure 3).

## 4. Discussion

Despite recent advances in the therapeutic approaches against ESC, such as surgery, chemotherapy, radiotherapy and CRT [6], the prognosis of patients remains substantially poor, as demonstrated by a 15–40% 5-year post-esophagectomy survival rate [7]. Therefore, there is an urgent need to develop a novel biomarker for ESC that has the potential to detect the cancer incidence at an early stage and predict the therapeutic response of the patients. In the present study, we found a significant reduction in the telomeric G-tail length in 147 patients with ESC, compared to that in 170 age-matched healthy controls. Our study also revealed that a shorter G-tail length was associated with an increased risk of ESC recurrence. Therefore, these results provide evidence that the G-tail length is a potential risk marker that reflects the physiological status of patients with ESC recurrence.

The process of the telomere shortening results in the genome destabilization and eventually leads to cellular senescence [25]. In addition to the telomere shortening, the G-tail shortening is a critical factor in the cellular senescence and aging. Therefore, dysfunction or the shortening of telomeres and the G-tail length is considered to cause age-related diseases [26]. Consistent with these findings, our previous studies reported that the G-tail shortening is associated with the onset and progression of several diseases, such as cardiovascular disease and breast cancer [22,23]. Since the LTL is not consistently associated with EAC [27], the effect of the LTL on ESC is debated. In this study, we found that the telomere and G-tail shortening was associated with aging in both controls and patients with ESC. However, the trend of the G-tail length shortening among patients with ESC over 50 years of age was more rapid, compared to the control participants. Considering that the incidence of ESC was significantly increased in individuals aged 50–69 years [28], these results suggest that the G-tail shortening is mainly caused by ESC incidence. To further investigate the significance of the G-tail shortening in ESC patients, it might be important to recruit healthy donors and ESC patients whose age is 50 or older.

The G-tail contains repetitive guanine bases, which are easily oxidized by reactive oxygen species (ROS), leading to the G-tail shortening [29]. Oxidative damage to DNA causes the formation of 8-hydroxy-2′-deoxyguanosine, which leads to G-T or G-A transversion mutations. Aging is accompanied by the increased ROS [30]. In addition, one of the risk factors for ESC is excessive alcohol consumption, which causes the generation of ROS [31,32]. Therefore, the accumulation of toxic ROS in patients with ESC might lead to the G-tail shortening. Considering the possibility that excessive alcohol consumption also induces the G-tail reduction in healthy individuals, for a more accurate evaluation of the G-tail shortening in ESC patients, it might be essential to examine the alcohol intake level of healthy individuals and ESC patients. 

A significant difference in the G-tail shortening was observed between the controls and preoperative patients with recurrent ESC. More importantly, a significant difference in the G-tail shortening was also observed between preoperative patients with and without ESC recurrence, while there was no significant difference of the LTL in the same compared groups. Although it is necessary to further investigate the correlation of the G-tail length between the pre- and post-recurrence patients, our study, for the first time, demonstrated that a shorter G-tail length was associated with an increased risk of ESC recurrence and it might be useful as a risk marker for the prediction of the ESC recurrence. To further investigate the meaning of the G-tail shortening in ESC recurrence, it might be quite important to conduct the study with a large sample size and long-term follow-up of ESC patients.

Carcinoembryonic antigen and squamous cell cancer antigen have been used as a tumor marker for ESC diagnosis [33]. Compared to these conventional markers, the benefits of the G-tail evaluation are non-invasive and easy-to-perform to evaluate the physiological status of ESC patients. Therefore, the combination with the G-tail evaluation and other ESC diagnostic markers might enable the prediction of ESC recurrence.

## 5. Conclusions

Our current study suggests that the evaluation of the G-tail length is useful for the prediction of ESC recurrence. Since circulating tumor cells (CTCs), secreted DNAs, and RNAs are considered to reflect the status of cancer patients [34,35], a combination of the G-tail evaluation and these molecules might lead to the development of more accurate and sensitive biomarkers for ESC patients.

## Figures and Tables

**Figure 1 jcm-11-07385-f001:**
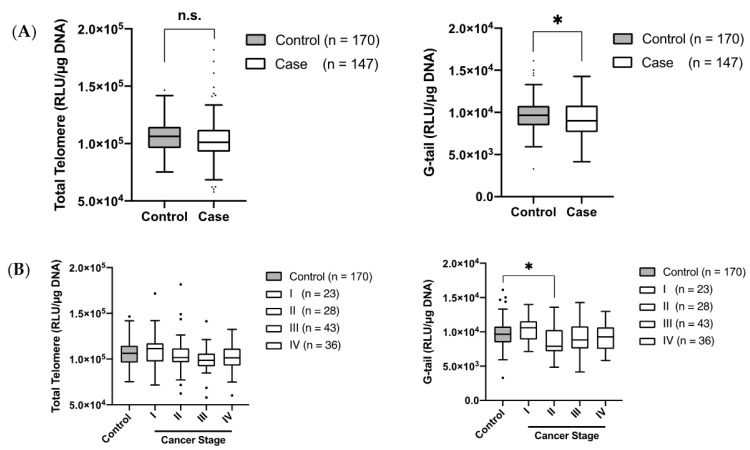
Comparison of the LTL/G-tail length in the controls and patients with ESC. (**A**) Box-and-whisker plot representing the comparison of the LTL (left panel) and the G-tail length (right panel) between the controls and patients with ESC. (**B**) Box-and-whiskers plot representing the comparison of the LTL (left panel) and G-tail length (right panel) between the controls and patients with ESC at different stages. Error bars indicate the standard error of the mean; n.s. non-significant, *p*-value > 0.05; * represents the significant difference (0.01 < *p* < 0.05).

**Figure 2 jcm-11-07385-f002:**
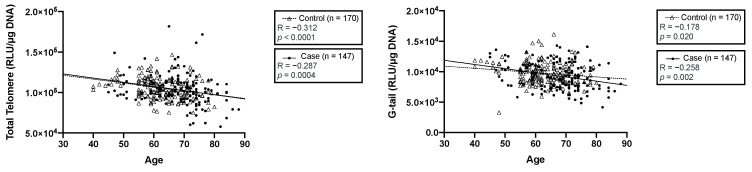
Correlation between aging and the LTL/G-tail length. The scatter plot represents the correlation between aging and the LTL (**left panel**) and aging and the G-tail length (**right panel**) in the controls and patients with ESC.

**Figure 3 jcm-11-07385-f003:**
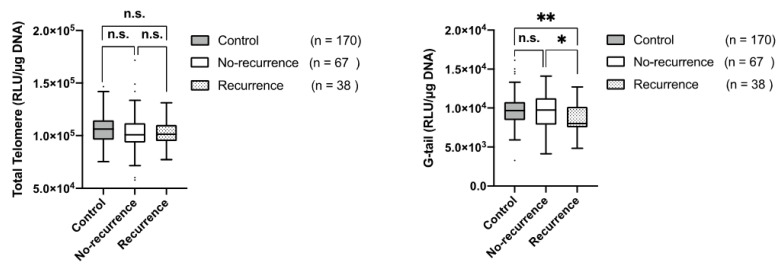
Association between the LTL/G-tail length and the possibility of recurrence. Box-and-whisker plot represents the comparison of the LTL (**left panel**) and the G-tail length (**right panel**) between the controls and preoperative patients with ESC with no recurrence or recurrence. Error bars indicate the standard error of the mean; n.s. represents non-significant, *p*-value > 0.05; * represents significant difference (0.01 < *p* < 0.05); ** represents significant difference (0.001 < *p* < 0.01).

**Table 1 jcm-11-07385-t001:** General characteristics of the study subjects.

Variable	Case (*n* = 147)	Control (*n* = 170)
Age	69.0 (63.0–74.0)	62.0 (58.0–68.3)
Gender		
Male	122 (83.0%)	88 (51.8%)
Female	25 (17.0%)	82 (48.2%)
Cancer type		
Squamous cell CA	123 (89.1%)	-
Adeno CA	10 (7.2%)	-
Basaloid squamous cell CA	2 (1.4%)	-
Neuroendocrine CA	1 (0.7%)	-
Small cell neuroendocrine CA	1 (0.7%)	-
Small cell CA	1 (0.7%)	-
Cancer stage		
I	23 (17.7%)	-
II	28 (21.5%)	-
III	43 (33.1%)	-
IV	36 (27.7%)	-
TNM stage (T category)		
1	29 (21.6%)	-
2	12 (9.0%)	-
3	73 (54.5%)	-
4a	5 (3.7%)	-
4b	15 (11.2%)	-
TNM stage (N category)		
0	52 (38.8%)	-
1	40 (29.9%)	-
2	34 (25.4%)	-
3	8 (6.0%)	-
TNM stage (M category)		
0	114 (85.1%)	-
1	20 (14.9%)	-
Therapeutic status	107	-
Preoperative		
Chemotherapy-only	11 (10.3%)	-
CRT-only	5 (4.7%)	-
Non-treated	79 (73.8%)	-
Postoperative		
Cancer-related surgery-only	7 (6.5%)	-
Chemotherapy	4 (3.7%)	-
CRT	1 (0.9%)	-
Recurrence status	114	-
Preoperative		
Non-recurrence	67 (58.5%)	-
Recurrence	38 (33.3%)	-
Postoperative		
Non-recurrence	5 (4.4%)	-
Recurrence	4 (3.5%)	-

For the non-normally distributed variables, the median (IQR), and categorical variables, the numbers of observations (%) in the category are given. -, not applicable; CA, carcinoma; CRT, chemoradiotherapy; IQR, interquartile range; TNM, tumor-node-metastasis.

## Data Availability

The data presented in this study are available upon request from the corresponding author. The data are not publicly available because they contain information that can compromise the privacy of the research participants.

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
