# Peer review of "Leukocyte Telomeric G-Tail Length Shortening Is Associated with Esophageal Cancer Recurrence"

_jcm, 2022, doi:10.3390/jcm11247385_

Round 1

Reviewer 1 Report

The authors identified a significant difference in G-tail length between healthy individuals (n=170) and patients with ESC (n=147). Surprisingly, the difference in telomere length was not significant between the healthy control and patient group. The manuscript is well written, although I question the novelty and authors claims that G-tail length could be a useful biomarker for ESC incidence and recurrence.

Major points:

·         Although the control group and patient group were “aged-matched”, they were not matched in gender and number. Could the authors use their data to show that there is no significant difference between gender and telomere length and re-analyze their data with equal numbers between the control and patient groups.

·         I doubt the validity of using G-tail length for ESC diagnosis, given that the only significant difference was between stage II patients. In addition, there was no difference in G-tail length in patients with non-reoccurrence ESC, suggesting that the test is only useful if you have already had ESC. Could the authors please elaborate on this point. Could they also compare the benefits of using G-tail length and other ESC diagnostic tests?

·         A significant difference in G-tail shortening was observed between preoperative patients with recurrent ESC compared to those with no recurrence. This led the authors to conclude that G-tail length might be useful for predicting ESC recurrence. Is there any way to test this theory with this data set? Do they have any pre- and post-reoccurrence samples from the same patients?

Reviewer 2 Report

This is an interesting study and a promising step toward the diagnosis of esophageal cancer. I have no major comments. A minor revision I suggest is to add a line or two mentioning the limitations of their study either in the conclusion or discussion part of the article. I recommend this manuscript be accepted after addressing the minor comment. 

Reviewer 3 Report

1. There are few grammatical, typos and syntax errors have been noted in the manuscript and it should be corrected.

2. Check the abbreviations throughout the manuscript and introduce the abbreviation when the full word appears the first time in the abstract and the remaining for the text and then use only the abbreviation (For example, chemoradiotherapy (CRT) etc.,). Make a word abbreviated in the article that is repeated at least three times in the text, not all words need to be abbreviated.

3. In materials and methods, the authors should include what are all the criterias are followed for the selection of healthy volunteers.

4. In materials and methods, the authors should clearly mentioned  how the inclusion and exclusion criterias are done in the present investigation.

5. The authors may improve the discussion of their work by focusing on the present findings and introducing data from other authors who also worked with the same or other studies with recent references.

6. In the conclusion seems to be in general. All conclusions must be convincing statements on what was found to be novel, impact based on the strong support of the data/results/discussion. And also, it is highly recommended to include limitation of the study and potential future research goals.

Round 2

Reviewer 1 Report

The Authors have addressed my concerns adequately. I still do not think these data suggest that G-tail length is associated with ESC incidence, as there was no difference in G-tail length between healthy individuals and patients with non-reoccurrence ESC (figure 3). I believe that the title should be changed to reflect this. I suggest the following as an alternative:

"Leukocyte Telomeric G-tail Length Shortening is Associated with Esophageal Cancer Recurrence"
